# Validation of performance of Spanish version of PURE-4 questionnaire for early identification of psoriatic arthritis after 1 year of follow-up in patients with psoriasis

R. Queiro-Silva[1]*, A. López-Ferrer[2], M. Ferran i Farrés[3], R. Rivera-Díaz[4],
D. Vidal-Sarro[5], L. Rodríguez Fernández-Freire[6], P. de la Cueva-Dobao[7],
J. Santos-Juanes[8], V. Rocamora-Duran[9], L. Gómez-Labrador[10], I. Belinchón-Romero[11],
on behalf of the PURE-4 Study Group¶

1 Division of Rheumatology, Hospital Universitario Central de Asturias, Oviedo, Spain, 2 Dermatology Department, Hospital de la Santa Creu i Sant Pau, Barcelona, Spain, 3 Dermatology Department, Hospital del Mar, Barcelona, Spain, 4 Dermatology Department, Hospital 12 de octubre, Universidad Complutense, Madrid, Spain, 5 Dermatology Department, Hospital de Sant Joan Despí Moisés Broggi, Barcelona, Spain, 6 Dermatology Department, Hospital Universitario Virgen del Rocío, Seville, Spain, 7 Dermatology Department, Hospital Universitario Infanta Leonor, Madrid, Spain, 8 Dermatology Department, Hospital Universitario Central de Asturias, Oviedo, Spain, 9 Dermatology Department, Hospital de Manacor, Palma de Mallorca, Spain, 10 Novartis Farmacéutica S.A., Barcelona, Spain 11 Dermatology Department, Hospital General Universitario Dr. Balmis de Alicante-ISABIAL, Universidad Miguel Hernández, Alicante, Spain,

☯ These authors contributed equally to this work.
¶ Membership of the PURE-4 Study Group is provided in the acknowledgments.
* rubenque7@yahoo.es

## Abstract

### Background and objective

Psoriatic arthritis (PsA) is an inflammatory, chronic and progressive musculoskeletal disease associated with psoriasis. The validation of the Spanish version of the Psoriatic arthritis UnclutteRed screening Evaluation (PURE-4) questionnaire has been published previously. The present analysis studied the performance of the PURE-4 questionnaire for the early detection of PsA one year before its diagnosis.

### Materials and methods

This was an observational multicenter study, including two cross-sectional assessments, with primary data collection in routine clinical practice in Spain. Adult patients with psoriasis and confirmed not having PsA diagnosis during Assessment I completed Assessment II by the rheumatologist one year (±2 months) after answering the PURE-4 questionnaire. This work presents the results of Assessment II, to confirm/rule out the presence of PsA in patients with psoriasis one year after PURE-4 answering. The analysis, involving the results from Assessment II, will evaluate the performance of the PURE-4 questionnaire for the early detection of potential PsA in terms of sensitivity and specificity.

**Data availability statement:** All relevant data are within the manuscript and its Supporting Information files.

**Funding:** This study was funded by Novartis Farmacéutica S.A.

**Competing interests:** I have read the journal's policy and the authors of this manuscript have the following competing interests: Rubén Queiro-Silva has received fees as a consultant, speaker, and project coordinator from the following pharmaceutical companies: AbbVie, Lilly, UCB, Janssen, Pfizer, Amgen, and Novartis, and has received grants from Janssen, Novartis, and AbbVie. Anna López-Ferrer has received fees for her participation as a member of scientific committees, consultant and/or speaker, and has received grants for research or participation in clinical trials from the following pharmaceutical companies (but not in relation to this manuscript): AbbVie, Almirall SA, Amgen, BMS, Boehringer Ingelheim, Janssen, LEO Pharma, MSD, Eli Lilly, Novartis, and UCB Pharma. Marta Ferran i Farrés has received fees as a consultant and/or speaker and has participated in clinical trials sponsored by the following pharmaceutical companies: Johnson & Johnson Innovative Medicine, Eli Lilly, Novartis, Pfizer-Wyeth, MSD, AbbVie, Celgene, BMS, Leo-Pharma, UCB, and Almirall SA. Raquel Rivera-Díaz has received fees as a consultant and/or speaker and has participated in clinical trials sponsored by the following pharmaceutical companies: AbbVie, Almirall SA, Amgen, Boehringer, BMS, Janssen Pharmaceuticals Inc, Eli Lilly, Leo-Pharma, Novartis, Pfizer-Wyeth, and UCB. David Vidal-Sarro has received fees as a consultant and/or speaker and has participated in clinical trials sponsored by the following pharmaceutical companies: AbbVie, Celgene, Eli Lilly, Janssen Pharmaceuticals Inc, Novartis, Gebro Pharma, Leo-Pharma, and UCB. Lourdes Rodríguez Fernández-Freire has received fees as a consultant and/or from the following pharmaceutical companies: AbbVie, Janssen Pharmaceuticals Inc, MSD, Pfizer-Wyeth, Novartis, Celgene, Almirall SA, Eli Lilly, and Leo-Pharma. Pablo de la Cueva-Dobao has received fees as a consultant and/or speaker and has participated in clinical trials sponsored by the following pharmaceutical companies: AbbVie, Almirall SA, Astellas

## Results

There were 219 evaluable patients, 56.2% male, and the mean (standard deviation [SD]) age was 46.8 (12.5) years. At one year, the PsA diagnosis was confirmed in 12 (5.5%) patients, representing 26.1% of the total number of patients with PsA diagnosed since the beginning of the study. The mean (SD) PURE-4 score was 2.4 (1.1) for patients with PsA and 1.2 (1.2) for patients without a diagnosis of PsA (p = 0.0016). The area under the receiver-operating characteristic (ROC) curve confirmed the good quality of the questionnaire (0.7618; 95% CI: 0.6530–0.8706; n = 217). PURE-4 showed a sensitivity of 75.0% and a specificity of 62.9%.

## Conclusions

The PURE-4 questionnaire offers good clinimetric capabilities for the early detection of PsA with a score ≥2. Of the total number of patients with PsA, one in four were detected one year after answering positively to the questionnaire, which would help to predict which patients are at high risk of developing PsA. The authors reinforce the recommendation to closely follow up with these patients.

## Introduction

Psoriatic arthritis (PsA) is an inflammatory, chronic and progressive musculoskeletal disease of recurrent course, with heterogeneous clinical manifestations and strongly associated with psoriasis, with a variable pattern of joint involvement [1]. According to data from the EPISER 2016 registry, the prevalence of PsA in the adult population in Spain is 0.58% [2], while the prevalence of psoriasis is 2.3% [3]. A study conducted in Canada estimated an annual incidence rate for PsA of 2.7% [4]. In 84% of individuals with psoriasis, the skin condition appears before the development of PsA [5].

PsA is a difficult diagnosis to make, and many cases go underdiagnosed; however, various studies place it between 15% and 40% of patients with psoriasis [5–7]. Delayed diagnosis contributes to the development of peripheral joint erosions and results in poor long-term physical function and health status, while an early diagnosis can help prevent or halt joint damage, thereby improving the clinical and radiological prognosis of the disease [6,8]. Although rheumatologists should ideally diagnose, treat, and monitor the course of PsA, as skin involvement usually precedes joint involvement in up to 84% of cases [9], in most cases, dermatologists are responsible for suspecting and screening for PsA. Dermatologists and primary care physicians must have simple screening tools at their disposal [1,10,11].

Although there are currently various tools for screening for PsA, they have not been culturally and linguistically adapted into Spanish, nor have they been validated through a structured development methodology within routine clinical practice in Spain [12]. The Psoriatic arthritis UnclutteRed screening Evaluation (PURE-4) questionnaire is a simple tool with only four items that evaluates signs suggestive of dactylitis, enthesitis, axial involvement, and peripheral involvement. It has recently been

Pharma, Biogen Inc., Boehringer Ingelheim, Celgene, Janssen Pharmaceuticals Inc, LEO Pharma, Eli Lilly, MSD, Novartis, Pfizer, and UCB. Jorge Santos-Juanes has received fees as a consultant and/or speaker from the following pharmaceutical companies: Novartis, Eli Lilly, Janssen Pharmaceuticals Inc, AbbVie, Amgen, and Sanofi. Vicenç Rocamora-Duran has received fees as a consultant and/or speaker from the following pharmaceutical companies: Janssen Pharmaceuticals Inc, Eli Lilly, AbbVie, Almirall SA, Amgen, and Novartis. Lara Gómez-Labrador is an employee of Novartis in Spain. Isabel Belinchón-Romero has received fees as a consultant and/or speaker and has participated in clinical trials sponsored by the following pharmaceutical companies whose therapeutic armamentarium includes drugs used to treat psoriasis: Janssen Pharmaceuticals Inc, Almirall SA, Lilly, AbbVie, Novartis, Celgene, Biogen, Amgen, Leo-Pharma, Pfizer-Wyeth, BMS, UCB, and MSD.

adapted linguistically and culturally for the Spanish population following the standardized methodology [11]. Following this adaptation, and with the aim of implementing the use of the questionnaire in clinical practice in Spain, it was validated and shown to be a sensitive (79.4%), specific (61.4%), and a feasible instrument for identifying patients with psoriasis at risk of developing PsA [13]. Finally, and as a continuation of this validation, this study analyzes for the first time the performance of the PURE-4 questionnaire as a tool for detecting patients at risk of developing PsA (one year after the questionnaire was completed).

## Materials and methods

### Study design

The cross-sectional, observational multicenter study was conducted in Spain, with primary data collection as per routine clinical practice in 19 public hospitals. Each hospital included a dermatology specialist and a rheumatology specialist who routinely assessed patients with psoriasis and PsA. All study visits were scheduled according to routine clinical practice. The study included adult (≥18 years) patients with psoriasis without a previous diagnosis of PsA who were consecutively enrolled based on clinical practice, as they met all the inclusion criteria and none of the exclusion criteria, and voluntarily agreed to participate [13]. More information about sites and study recruitment data is presented in S1 Table.

### Study assessments

The study consisted of two cross-sectional assessments (Fig 1):

- Assessment I: From December 22, 2020, to August 17, 2021, patients were assessed by a dermatologist and completed two self-administered versions (in print and online) of the PURE-4 questionnaire. Thereafter, the rheumatologist, blinded to the PURE-4 results, assessed the presence/absence of PsA, being the reference to determine the performance of the PURE-4 questionnaire [13].

- Assessment II: Until November 17, 2022, psoriasis patients without a confirmed diagnosis of psoriatic arthritis at the initial assessment (Assessment I) were re-evaluated by a rheumatologist approximately one year later (±2 months) to determine the presence of PsA based on clinical judgment.

The results of Assessment I allowed for the validation of the Spanish version of the PURE-4 questionnaire and were previously reported [13]. This paper presents the results of Assessment II, the aim of which was to analyze the performance of the PURE-4 questionnaire as a tool for the early detection of PsA one year after having answered it (in Assessment I) to confirm/rule out the presence of PsA in patients with psoriasis and patients without a diagnosis of PsA after Assessment I.

The study variables included, but were not limited to, the sociodemographic and clinical data of the patients [13]. In Assessment II, the sensitivity and specificity of the PURE-4 questionnaire were studied to confirm the presence/absence of PsA in patients without a diagnosis of PsA after Assessment I one year after answering the PURE-4 questionnaire.

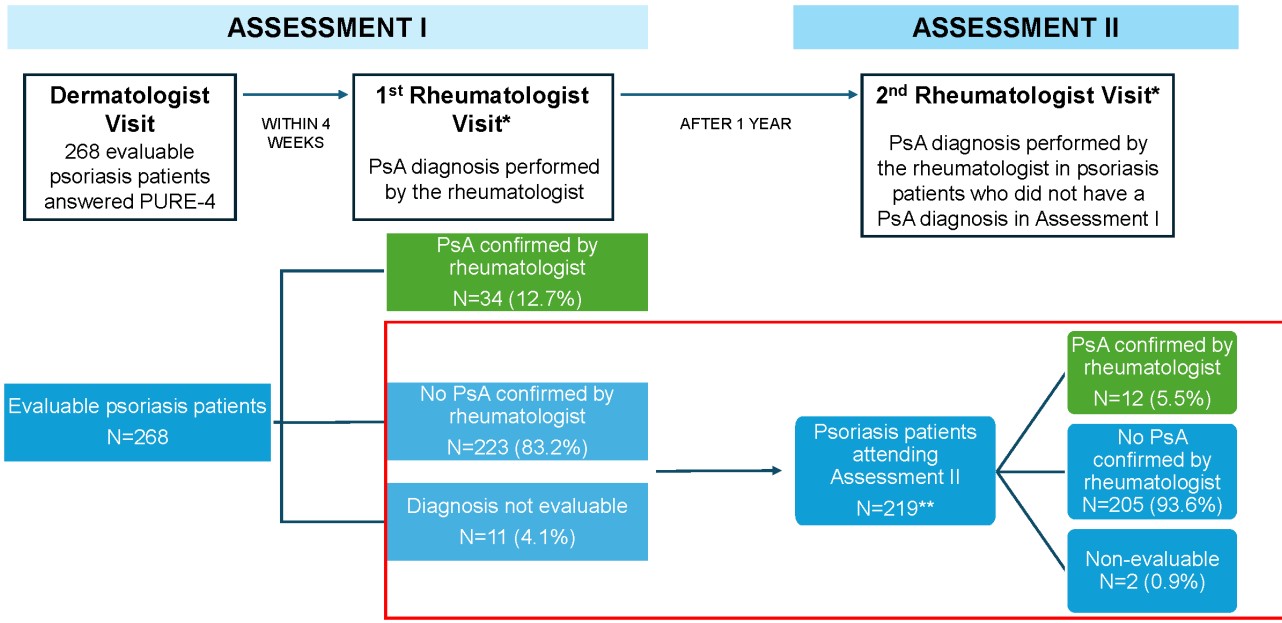

**Fig 1. Study design: This paper presents the results of Assessment II (boxed in red).** *Rheumatologist was blinded during the two PsA assessments to the PURE-4 results. **15 patients did not attend the second visit to the rheumatologist. PsA, psoriatic arthritis.

The reporting of this study conforms to STROBE guidelines [14].

## Statistical analysis

Two statistical analyses were planned, the first of which was previously reported [13]. The second and final analysis involved the results of Assessment II to assess the performance of the PURE-4 questionnaire for the early detection of potential PsA one year after completing the PURE-4 questionnaire, in terms of sensitivity, specificity, positive predictive value (PPV), negative predictive value (NPV), and the percentage of patients correctly classified, with a 95% confidence interval (95% CI).

The area under the receiver–operating characteristic (ROC) curve with its 95% CI was calculated to assess the capacity or quality of the PURE-4 questionnaire to identify patients with PsA, considering the final diagnostic confirmation performed by the rheumatologist.

To select optimal cut-off score, the Youden index was calculated. The cut-off score associated with the highest index is considered to indicate the optimal cut-off point. Youden index is the maximum vertical distance or difference between the ROC curve and the diagonal or chance line and occurs at the cut point that optimizes the biomarker's differentiating ability when equal weight is given to sensitivity and specificity.

In Assessment II, for a specificity ≥0.80 with a precision of 5.2 percentage points and a significance level of 0.05, a subsample of 227 patients without a diagnosis of PsA would be needed. Assuming a 10% loss rate, for the validation of the questionnaire in terms of sensitivity and specificity for the early detection of PsA, the number of patients required would be 205.

The PURE-4 questionnaire scores were obtained only in Assessment I and were compared with the presence or absence of PsA according to the clinical judgment of the rheumatologist in Assessment II (one year later).

A descriptive analysis of the sociodemographic and clinical characteristics was performed. Psoriasis Area and Severity Index (PASI), body surface area (BSA), Physician Global Assessment (PGA), Dermatology Life Quality Index (DLQI), and

CASPAR criteria were also described. When the rheumatologist considered necessary, dactylitis, enthesitis, inflammatory back and peripheral joint pain were assessed by complementary tests, according to clinical practice, and also described.

Quantitative data were described using valid N, missing N, mean, standard deviations (SDs), extreme values and quartiles. Categorical variables were described using frequencies and percentages in relation to class level. In both cases, the number of observations and the number of missing data were specified. No imputation of missing values was performed for any variable. Comparisons between patients with and without a diagnosis of PsA were performed using T-tests, Mann-Whitney and Chi-square tests, without taking into account the subgroup with a non-evaluable diagnosis and considering a statistical significance level of 0.05. The statistical software Statistical Analysis System (SAS) Enterprise Guide version 7.15 was used.

### Ethical statement

The study was performed in accordance with the ethical principles of the Declaration of Helsinki and the guidelines specified in Order SAS/3470/2009 of the Spanish Agency of Medicines and Medical Devices (AEMPS). The protocol, informed consent, and patient information were approved by the central Ethics Committee (EC) of the Hospital de Bellvitge, as well as by all the ECs and autonomous communities of the participating hospitals, as required. All patients signed written informed consent before enrollment in the study.

### Results

In Assessment I, a total of 283 patients were enrolled, of whom 268 (94.7%) were evaluable. Of note, 34 (12.7%) were diagnosed with PsA [13]. In Assessment II, 219 patients with psoriasis and no diagnosis of PsA were included and made a second visit to the rheumatologist.

In Assessment II (n = 219), the diagnosis of PsA was confirmed in 12 (5.5%) patients. Taking into account both study assessments (n = 253), a total of 46 patients (18.2%) with PsA were identified [13].

It was not possible to determine whether or not 15 of the 268 evaluable patients in the study had PsA. Hence, it is considered that in 253 patients (94.4%), the diagnosis of PsA could be confirmed or ruled out.

#### Baseline characteristics of patients included in Assessment II

At Assessment II, the mean (SD) age was 43.4 (12.5) years for patients with a diagnosis of PsA and 47.0 (12.5) years for those without a diagnosis of PsA. Of the patients with and without a diagnosis of PsA, 66.7% and 56.1%, respectively, were male. The mean time from the first diagnosis of psoriasis to inclusion in the study was longer in patients with a diagnosis of PsA than in those without PsA (21.0 vs. 18.7 years). Overall, patients with PsA had a higher percentage of dactylitis (8.3%), enthesitis (16.7%), inflammatory back pain (16.7%), and peripheral joint pain (58.3%) than patients without PsA (0%, 0%, 1.0%, and 3.4%, respectively) (Table 1).

A total of 88.1% of patients included in Assessment II were on treatment for psoriasis, mainly with biologic drugs (46.1%) (Fig 2).

More information on the overall results of the study is presented in S2 Table and S1 Fig.

#### Assessment II: Performance of the PURE-4 questionnaire in early PsA identification

Throughout the study, 78.2% of patients diagnosed with PsA scored ≥2 on the PURE-4 questionnaire at baseline, with 79.4% being those diagnosed in Assessment I [13] and 75.0% in Assessment II (Fig 3).

In Assessment I, the rheumatologist was unable to confirm the diagnosis of PsA in 14 patients, responding "currently no, but presents signs and/or symptoms suggestive of early stage PsA." Two of these 14 patients did not attend the second visit to the rheumatologist, and of those patients who attended, 4 (33.3%) were diagnosed with PsA.

**Table 1. Baseline clinical and sociodemographic characteristics of evaluable psoriasis patients who attended Assessment II (variables collected in Assessment I).**

| Characteristics | With a diagnosis of PsA (n=12) | Without a diagnosis of PsA (n=205) | Non-evaluable (n=2) | Total (n=219) | P-value |
|---|---|---|---|---|---|
| Age, years, mean (SD) | 43.4 (12.5) | 47.0 (12.5) | 39.0 (19.8) | 46.8 (12.5) | 0.2913 |
| Male, n (%) | 8 (66.7) | 115 (56.1) | 0 | 123 (56.2) | 0.4727 |
| Years from the diagnosis of psoriasis, mean (SD) | 21.0 (15.2) | 18.7 (12.8) | 13.1 (9.8) | 18.7 (12.8) | 0.6564 |
| Special sites of psoriasis, n (%) | 8 (66.7) | 95 (46.3) | 1 (50.0) | 104 (47.5) | 0.1705 |
| Patients on treatment for psoriasis, n (%) | 9 (75.0) | 182 (88.8) | 2 (100.0) | 193 (88.1) | 0.1531 |
| PASI, mean (SD) | 9.4 (6.4) | 6.8 (5.0) | 9.5 (3.5) | 7.0 (5.1) | 0.2382 |
| Mild (PASI <7), n (%) | 3 (25.0) | 93 (45.4) | 0 | 96 (43.8) | 0.1674 |
| Moderate/severe (PASI ≥7), n (%) | 9 (75.0) | 112 (54.6) | 2 (100.0) | 123 (56.2) | |
| BSA, mean (SD) | 12.5 (9.7) | 8.2 (8.1) | 5.0 (-) | 8.5 (8.2) | 0.1468 |
| Valid n | 11 | 165 | 1 | 177 | |
| PGA | | | | | |
| Valid n | 10 | 158 | 1 | 169 | |
| 0. Clear: No signs of psoriasis (post-inflammatory hyperpigmentation may be present), n (%) | 1 (10.0) | 14 (8.9) | 0 | 15 (8.9) | 0.3835 |
| 1. Nearly clear, minimal: Minimal plaque elevation, scaling and/or erythema, n (%) | 1 (10.0) | 35 (22.2) | 0 | 36 (21.3) | |
| 2. Mild: Mild plaque elevation, scaling and/or erythema, n (%) | 1 (10.0) | 46 (29.1) | 1 (100.0) | 48 (28.4) | |
| 3. Moderate: Moderate plaque elevation, erythema and/or scaling, n (%) | 5 (50.0) | 49 (31.0) | 0 | 54 (32.0) | |
| 4. Severe: Very marked plaque elevation, erythema and/or scaling, n (%) | 2 (20.0) | 14 (8.9) | 0 | 16 (9.5) | |
| DLQI, mean (SD) | 8.4 (9.2) | 7.6 (6.7) | 20.0 (-) | 7.8 (6.9) | 0.9287 |
| Valid n | 9 | 118 | 1 | 128 | |
| Inflammatory signs and imaging tests (results after complementary tests [all patients]) | | | | | |
| Presence of dactylitis, n (%) | 1 (8.3) | 0 | 0 | 1 (0.5) | 0.0002 |
| Presence of enthesitis, n (%) | 2 (16.7) | 0 | 0 | 2 (0.9) | <0.0001 |
| Presence of inflammatory back pain, n (%) | 2 (16.7) | 2 (1.0) | 0 | 4 (1.8) | <0.0001 |
| Presence of peripheral joint pain, n (%) | 7 (58.3) | 7 (3.4) | 0 | 14 (6.4) | <0.0001 |
| Meets CASPAR criteria, n (%) | 7 (58.3) | 0 | 0 | 7 (3.2) | <0.0001 |
| CASPAR, mean (SD) | 3.3 (0.5) | – | – | 3.3 (0.5) | |

BSA, body surface area; CASPAR, ClASsification criteria for Psoriatic Arthritis; DLQI, Dermatology Life Quality Index; n, number of patients (unless otherwise specified); PASI, Psoriasis Area and Severity Index; PGA, Physician Global Assessment; PsA, psoriatic arthritis; SD, standard deviation.

In Assessment II, the mean (SD) PURE-4 score was 2.4 (1.1) among patients with PsA and 1.2 (1.2) among patients without a diagnosis of PsA (Table 2). Regarding the questionnaire items, patients with PsA showed greater presence of all of the items than patients without PsA, with peripheral joint pain with swelling before the age of 50 years (item 4) being the most prevalent symptom, and inflammatory heel pain (item 2) being the only one that did not show significant differences between patients with and without PsA (Fig 4).

Items 2 and 4 of the questionnaire showed the greatest difference among patients with a diagnosis of PsA in the two assessments (S2 Fig).

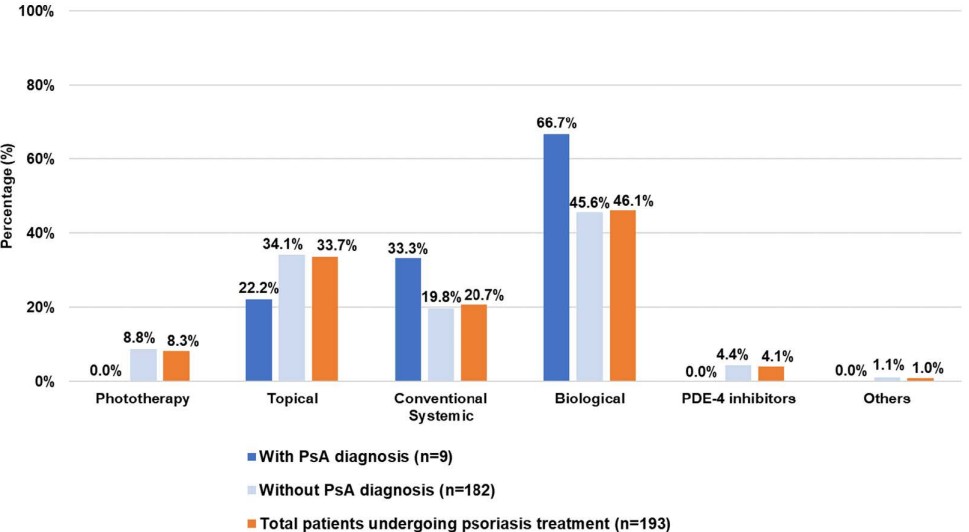

**Fig 2. Baseline treatment of patients undergoing psoriasis treatment included in Assessment II.** PDE-4, phosphodiesterase 4; PsA, psoriatic arthritis.

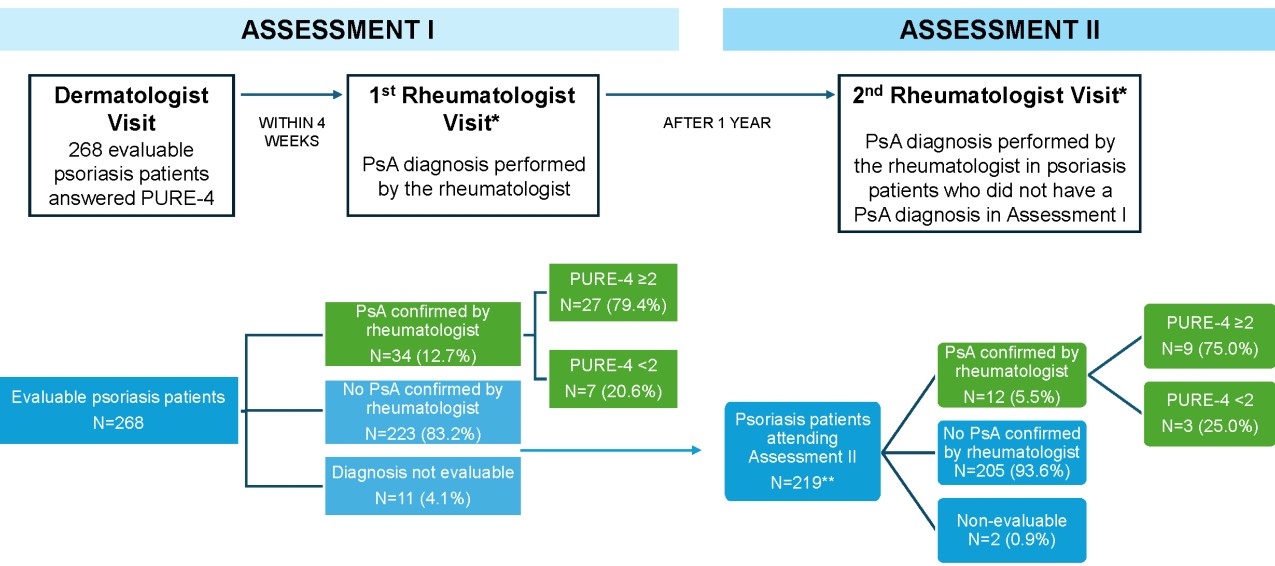

**Fig 3. Flowchart of study population (Assessment I and Assessment II).** *Rheumatologist was blinded during the two PsA assessments to the PURE-4 results. **15 patients did not attend the second visit to the rheumatologist. PsA, psoriatic arthritis.

## Sensitivity and specificity

The sensitivity and specificity of the questionnaire were evaluated considering those patients with psoriasis in whom the presence/absence of PsA was confirmed by a rheumatologist in Assessment II (n = 217), showing that the area under the ROC curve was 0.7618 (95% CI: 0.6530–0.8706), indicating the good performance of the questionnaire in the identification of early PsA (Fig 5).

**Table 2. Results of PURE-4 questionnaire in Assessment II.**

| Variable | With a diagnosis of PsA (n = 12) | Without a diagnosis of PsA (n = 205) | Non-evaluable (n = 2) | Total (n = 219) | P-value |
|---|---|---|---|---|---|
| PURE-4 score (0–4), mean (SD) | 2.4 (1.1) | 1.2 (1.2) | 3.0 (1.4) | 1.3 (1.3) | 0.0016 |
| Number of positive items in PURE-4 questionnaire | | | | | |
| 0, n (%) | 0 | 77 (37.6) | 0 | 77 (35.2) | 0.0342 |
| 1, n (%) | 3 (25.0) | 52 (25.4) | 0 | 55 (25.1) | |
| 2, n (%) | 3 (25.0) | 38 (18.5) | 1 (50.0) | 42 (19.2) | |
| 3, n (%) | 4 (33.3) | 27 (13.2) | 0 | 31 (14.2) | |
| 4, n (%) | 2 (16.7) | 11 (5.4) | 1 (50.0) | 14 (6.4) | |

n, number of patients; PsA, psoriatic arthritis; PURE-4, Psoriatic arthritis UnclutteRed screening Evaluation; SD, standard deviation.

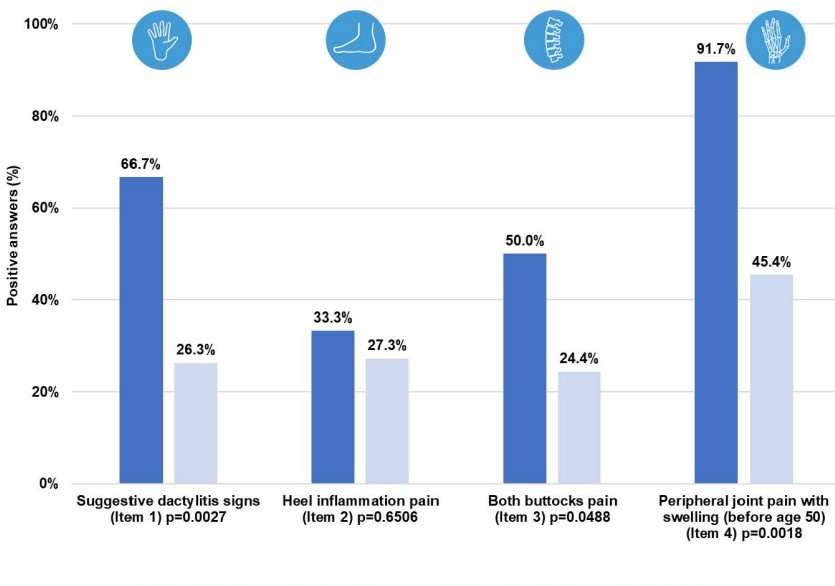

**Fig 4. Results of PURE-4 questionnaire by item in Assessment II.** n, number of patients; PsA, psoriatic arthritis.

At one-year follow-up after answering the PURE-4 questionnaire, sensitivity was 75.0% and specificity 62.9%. In 63.6% of the cases, the PURE-4 questionnaire classified the patients in the same way as the rheumatologist, with an NPV of 97.7% (Table 3).

Using the Youden index, it was determined, as in Assessment I, that a score ≥2 was the optimal cut-off point that allowed screening for the presence of potential PsA [13]. In Assessment II, the Youden index was very similar between the scores of 1 and 2 but, if score 1 was chosen, the specificity was much lower, with only 37.6% of patients who did not actually have a diagnosis of PsA being classified as not having a diagnosis of PsA (Table 3).

## Discussion

The identification of the patients with psoriasis at an increased risk of developing PsA, together with the early diagnosis of PsA, is of considerable scientific and clinical interest, as early diagnosis and treatment of PsA have been shown to delay

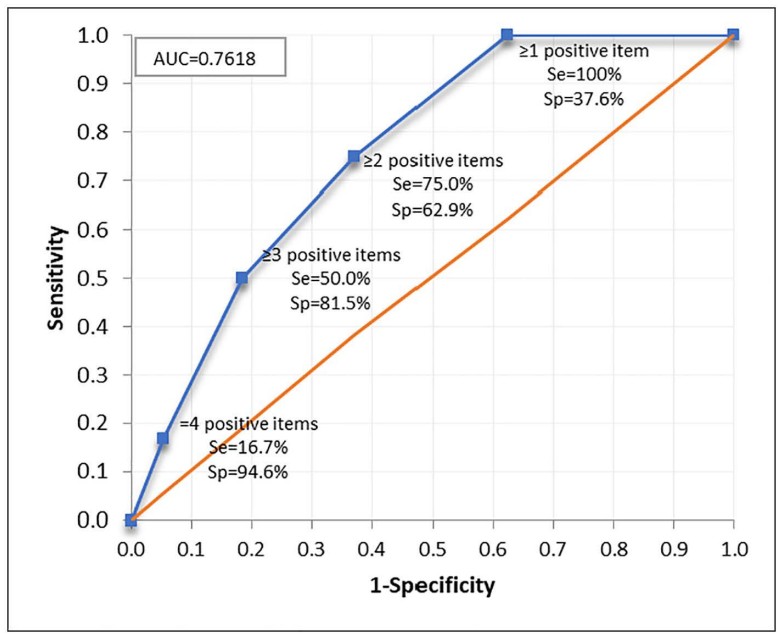

**Fig 5. Area under ROC curve in Assessment II.** AUC, area under the curve; ROC, receiver-operating characteristic; Se, sensitivity; Sp, specificity.

**Table 3. Sensitivity and specificity results for Assessment II.**

| PURE-4 cut-off points | Sensitivity (95% CI) | Specificity (95% CI) | PPV (95% CI) | NPV (95% CI) | % correct classification (95% CI) | Youden index |
|---|---|---|---|---|---|---|
| ≥1 | 100 (75.7–100) | 37.6 (31.2–44.4) | 8.6 (5.0–14.4) | 100 (95.2–100) | 41.0 (34.7–47.7) | 0.375 |
| ≥2 | 75.0 (46.8–91.1) | 62.9 (56.1–69.2) | 10.6 (5.7–18.9) | 97.7 (93.5–99.2) | 63.6 (57.0–69.7) | 0.379 |
| ≥3 | 50.0 (25.4–74.6) | 81.5 (75.6–86.2) | 13.6 (6.4–26.7) | 96.5 (92.6–98.4) | 79.7 (73.9–84.5) | 0.314 |
| =4 | 16.7 (4.7–44.8) | 94.6 (90.6–97.0) | 15.4 (4.3–42.2) | 95.1 (91.2–97.3) | 90.3 (85.7–93.6) | 0.113 |

95% CI, 95% confidence interval; NPV, negative predictive value; PPV, positive predictive value; PURE-4, Psoriatic arthritis UnclutteRed screening Evaluation.

and/or prevent permanent disability [15,16]. The role of dermatologists is crucial for the early detection and prompt referral of the patient with psoriasis at risk of PsA or with suspected PsA, and ultimately, early intervention by rheumatologists to prevent the progression of PsA, as well as the associated disability and associated comorbidities [1,16].

Recently, a group of the European Alliance of Associations for Rheumatology (EULAR) developed a guideline to define the suspected features of psoriasis progression to PsA and the clinical management of these patients [15]. This guideline includes three relevant stages for the progression of PsA: patients with psoriasis at increased risk of developing PsA, subclinical PsA, and clinical PsA [15]. All people with psoriasis are at risk of developing PsA, highlighting arthralgia as a key element of subclinical PsA that can be used as a possible short-term predictor of the development of PsA [15]. The present study was designed to validate the PURE-4 questionnaire for the early screening for PsA in patients with psoriasis

receiving routine dermatologic care in Spain (assessment I) [13], and to validate, for the first time, its potential as a tool for the detection of early PsA (Assessment II) one year after answering the PURE-4 questionnaire, demonstrating its good screening properties with an optimal cut-off point of ≥2. Furthermore, peripheral joint pain with swelling before the age of 50 (item 4) was observed to be the most prevalent symptom, in line with what EULAR indicates for the detection of sub-clinical PsA [15].

Some studies show that more than half of patients with PsA are diagnosed with years of delay [17], which is especially significant in Spain, with a mean of 4.0 ± 1.4 years [18]. Therefore, it is imperative to include a PsA screening tool that enables early detection in patients with psoriasis in clinical practice.

In addition to the points to be taken into account recommended by the EULAR group [15], the capacity for early detection of PsA was evaluated with the Early Arthritis for Psoriatic Patients (EARP) questionnaire [19], which concluded that it was effective with a specificity of 62%. However, a subsequent external validation of the EARP questionnaire showed a particularly low specificity (34%) [20]. Taking these results into account, PURE-4 seems to be a suitable screening questionnaire to detect PsA early, with a score ≥2 as optimal cut-off point, and both for its sensitivity (75.0%) and specificity (62.9%), which is especially relevant for a potentially disabling disease where the musculoskeletal health of patients is not only affected by aging, but also by the persistent increase or chronification of the inflammatory state [21,22]. Audureau E, et al, in a publication whose objective was to describe the development and internal validation of the PURE-4 questionnaire in a single center (not in a multicenter manner as in the case of the present manuscript) and which did not validate the performance of the tool 1 year before the diagnosis of PsA, obtained results aligned with our results, although slightly higher, where development and first validation of the PURE-4 scale showed a sensitivity of 85.7% and a specificity of 83.6% at the threshold of ≥1 positive item [12].

In these patients, early diagnosis can lead to prompt medical treatment and early lifestyle modifications, with a potential role in preventing joint deterioration, pain management, and quality of life [15,21,22]. Furthermore, although not analyzed in the present study, it is worth noting that some studies show a significant decrease in the risk of developing PsA in patients with psoriasis who receive biological treatments, concluding that the use of biological drugs should be considered at an earlier stage of treatment in patients who have significant risk factors for PsA [23,24].

The cumulative incidence of PsA in psoriasis patients ranges from 5.1% at 20 years to 20.5% at 30 years after the onset of psoriasis, being the annual incidence rate of PsA of 2.7% (95% CI: 2.1–3.6) in Canada, while the annual incidence rate of PsA in psoriasis patients without baseline arthritis was 1.87% in a prospective cohort study [4]. In the present study, an additional 5.5% of patients with PsA were identified one year after answering the PURE-4 questionnaire, in line with previous studies [19,25]. These percentages are higher than those detected in a previous epidemiologic survey, with around 10% of patients after a decade of follow-up [26].

The usefulness of the PURE-4 questionnaire in identifying PsA, even early, could be explained, in part, by its specific PsA items, including, among others, the age component (peripheral joint pain with swelling in patients <50 years), and it is one of the two questionnaires that include bilateral buttock pain [27].

The good performance of the PURE-4 questionnaire in its Spanish version in terms of sensitivity and specificity [13], as well as the fact that it includes four key questions on the identification of PsA, makes this questionnaire a tool that is easy to apply and viable in routine clinical practice considering the optimal cut-off point of ≥2 [27]. Furthermore, it is a quick questionnaire to use, and considering its high sensitivity, it could be useful to encourage its use at different time points (e.g., annually or in case of clinical doubt) to increase the identification of possible cases of PsA.

This study has some limitations inherent to the methodology used. The fact that the PURE-4 questionnaire was completed only at the beginning of the study (Assessment I) did not allow us to extend the initial objectives of the study by also assessing sensitivity to change or test-retest reliability. Consequently, administering the questionnaire only once limits the ability to evaluate the reproducibility of the score over time. This is the first time the performance of the PURE-4 questionnaire has been tested for detecting PsA with a one-year interval between its administration and diagnosis. This

fact is significant because it is recommended to screen psoriasis patients once a year (ideally, every six months), and the PURE-4 serves as a screening aid for PsA, although it does not absolve the responsibility of evaluating the patient, including their risk factors. Finally, given the small number of PsA cases at one year (n = 12), 95% CI are wide, and results should be interpreted with caution.

## Conclusions

The PURE-4 screening questionnaire has been shown to be a sensitive, specific, and feasible instrument to identify the risk of developing PsA in patients with psoriasis and, therefore, valid for application in Spanish dermatologic clinical practice. Its implementation is expected to reduce the rates of underdiagnosis of PsA by allowing dermatologists to refer patients early for early rheumatologic care. The data collected in our study allowed us to detect 46 (18.2%) patients with a diagnosis of PsA in one year: 34 (73.9%) at baseline and 12 (26.1%) after one year, which reinforces the recommendation to assess the presence of PsA annually [11], as well as to identify those patients at a higher risk of developing PsA for closer follow-up, as this could help to prevent irreversible joint damage in the context of this disease.

## Supporting information

**S1 Table. Sites and study recruitment data.** FPFV, First Patient First Visit; LPLV, Last Patient Last Visit; SIV, Site Initiation Visit.
(DOCX)

**S2 Table. Baseline clinical and sociodemographic characteristics of all evaluable psoriasis patients enrolled in Assessment I and Assessment II (variables collected in Assessment I).** BSA, body surface area; DLQI, Dermatology Life Quality Index; n, number of patients (unless otherwise specified); PASI, Psoriasis Area and Severity Index; PGA, Physician Global Assessment; PsA, psoriatic arthritis; SD, standard deviation.
(DOCX)

**S1 Fig. Baseline treatment of patients with psoriasis enrolled in study\*.** *Only those patients for whom treatment was indicated are included (treatment was not indicated in 29 patients, of whom 6 were diagnosed with PsA). **Total patients on treatment for psoriasis does not include 2 patients in whom the diagnosis of PsA was not evaluable. PDE-4, phosphodiesterase 4; PsA, psoriatic arthritis.
(TIFF)

**S2 Fig. Comparative analysis of PURE-4 questionnaire scores by item in patients with PsA at Assessment I and Assessment II.** n, number of patients; PsA, psoriatic arthritis.
(TIF)

## Acknowledgments

Membership of the PURE-4 Study Group: A. Pérez, J. Mollet, C. Pujol, R. Iglesias, M.I. Rodríguez, N. Jiménez, S. Pérez, P. Herranz, M. García, O. González, V. Jovani, D. Reina, A.M. Laiz, E. Beltrán, B. Joven, M.T. Navío, E. Rubio, R. Hernández, A. Erra, E. Vicens, F. Maceiras, M. Rodríguez, M. Valero, M.L. García, E. De Miguel, A. Genaro, C. Lerín, A. Aragón.

We would like to thank IQVIA, Carmen Barrull, Neus Canal, and Mónica Sarmiento for providing medical editorial support with this paper.

## Author contributions

**Conceptualization:** Rubén Queiro, I. Belinchón-Romero.

**Investigation:** Rubén Queiro, A. López-Ferrer, M. Ferran i Farrés, R. Rivera-Díaz, D. Vidal-Sarro, L. Rodríguez Fernández-Freire, P. de la Cueva-Dobao, J. Santos-Juanes, V. Rocamora-Duran, I. Belinchón-Romero.

**Methodology:** Rubén Queiro, I. Belinchón-Romero.

**Supervision:** Rubén Queiro, I. Belinchón-Romero.

**Writing – review & editing:** Rubén Queiro, A. López-Ferrer, M. Ferran i Farrés, R. Rivera-Díaz, D. Vidal-Sarro, L. Rodríguez Fernández-Freire, P. de la Cueva-Dobao, J. Santos-Juanes, V. Rocamora-Duran, L. Gómez-Labrador, I. Belinchón-Romero.

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
