## [Editor Report · Decision Letter 0]

15 Oct 2025

Dear Dr. Queiro,

Thank you for submitting your manuscript to PLOS ONE. After careful consideration, we feel that it has merit but does not fully meet PLOS ONE’s publication criteria as it currently stands. Therefore, we invite you to submit a revised version of the manuscript that addresses the points raised during the review process.

**ACADEMIC EDITOR:  Title and Abstract**

- Briefly describe psoriatic arthritis as a disease.

- Briefly give the key findings of your study. Include key numeric data (including confidence intervals or *p*  values)

**Introduction**

- Briefly describe psoriatic arthritis as a disease.

- The gaps in knowledge and rationale for the study need to be mentioned.

**Methods**

- Please outline this section following the STROBE guidelines.

- You need to state in the Methods section that you have followed relevant EQUATOR guidelines. e.g., STROBE: “The reporting of this study conforms to STROBE guidelines”. (Insert new reference number, Note, you will probably need to renumber your references after this addition).

- Need subheading for more clarity

- How were the patients selected (e.g., consecutively, randomly, or selectively)?

- Describe how the study size was arrived at.

**Discussion**

- In the grand scheme of things, please attempt to summarize key findings in relation to the study's objectives.

- Have you compared your results with relevant previous papers, and cited those papers?

- Have you discussed the relevance and novelty of your study and what it adds to literature?

- Ensure there are the following Declarations sections at the end of your manuscript: Acknowledgements, Author contributions, Funding, Availability of data and materials statement, competing interests, Ethics approval and consent to participate.

We look forward to receiving your revised manuscript.

Kind regards,

Wesam Gouda, MD,PhD

Academic Editor

PLOS ONE

Journal Requirements:

Reviewers' comments:

Reviewer's Responses to Questions

**Comments to the Author**

Reviewer #1: (No Response)

Reviewer #2: (No Response)

Reviewer #3: (No Response)

2. Is the manuscript technically sound, and do the data support the conclusions?

Reviewer #1: Yes

Reviewer #2: Partly

Reviewer #3: (No Response)

3. Has the statistical analysis been performed appropriately and rigorously?

Reviewer #1: Yes

Reviewer #2: Yes

Reviewer #3: (No Response)

4. Have the authors made all data underlying the findings in their manuscript fully available?

Reviewer #1: No

Reviewer #2: No

Reviewer #3: (No Response)

5. Is the manuscript presented in an intelligible fashion and written in standard English?

Reviewer #1: Yes

Reviewer #2: Yes

Reviewer #3: (No Response)

Reviewer #1: Your study provides valuable evidence on the potential of the PURE-4 questionnaire to identify psoriasis patients at increased risk of developing PsA. However, several aspects require revision to strengthen the manuscript:

1. Statistical power: Please emphasize the small number of PsA cases (n=12 at one year) and interpret sensitivity/specificity estimates with caution, highlighting the wide confidence intervals.

2. Treatment confounding: Discuss more explicitly how systemic and biologic therapies might have influenced PsA onset and questionnaire performance.

3. Comparison with existing tools: Include a discussion of how PURE-4 compares with other screening questionnaires (e.g., EARP), addressing advantages and limitations.

4. Study design limitation: Acknowledge that administering the questionnaire only once precludes assessment of reproducibility or changes in score over time.

Addressing these points will improve the clarity, robustness, and overall impact of your work.

Reviewer #2: The study objectives are interesting and relevant for the early detection of psoriatic arthritis. The article is carefully written and methodology is robust. I have provided my comments below.

Abstract:

1. Please restructure the ‘materials and methods’ part in the abstract in order to clarify to the readers that the patients with psoriasis (without previous diagnosis of PsA) from Assessment I took part in Assessment II. Please distinguish which cross-sectional studies the assessments are related to. Information on the sensitivity and specificity analysis will improve the results section further.

Introduction:

2. As the validation study of the 'PURE-4 scale' was conducted in a sample from France between 2012 and 2014, the following sentence needs to be updated: “Although there are currently various tools for screening for PsA, neither a development methodology nor validation in routine clinical practice in Spain has been carried out at all times [11].” I understand that you indicated its necessity for the Spanish sample. Please make it clear for the reader. (Page 4, line 79)

3. In the Introduction part, in addition to the prevalence data, please add a short description on the disease and how early detection helps in reducing disease burden. I recommend expanding the literature review a little bit and include more recent studies on this topic.

4. I also recommend shortly reviewing the previous studies and describing a clear research gap.

Methods:

5. Which software was used to calculate the performance of the PURE-4 questionnaire for the early detection of potential PsA in terms of sensitivity, specificity, positive predictive value, negative predictive value? (Page 6, line 132). Discuss in brief what the term ‘area under the receiver-operating characteristic (ROC) curve’ indicates about the quality of the questionnaire.

6. Please refer to the Youden index and describe it briefly.

Results:

7. The numbers presented in the results section (Page 7, line 165) is different from what is presented in Figure 1. For example, the figure shows that Dx: no PsA (n = 223) and Dx: not evaluable (n= 11) participated in Assessment II. Shouldn’t be it 234 instead of 219? Please present identical values and check for any mistakes. In figure 3, please present the information on the participants who were not evaluable.

8. There are differences in the values between Table 1 and S2 Table. Please check for any mistakes. If necessary, please update the title of the tables.

9. How Dactylitis, Enthesitis, inflammatory back pain, peripheral joint pain, DLQI, etc were evaluated? Please describe in short in the methods section.

Discussion:

10. What were the sensitivity and specificity of the original validation study for the PURE-4 scale? Please compare with your results.

11. Please discuss which relevant questions of the questionnaire were answered one year before the PsA were identified during assessment II. How does it determine the performance of PURE-4 questionnaire for early detection of PsA? If Figure 4 includes data only from assessment II (e.g., 91.7% participants with PsA diagnosis had peripheral joint pain with swelling), how can it be stated that it has assisted in early detection of PsA if it is done during the same assessment? You presented related results (both from two assessments) in S2 Fig. Please update the discussion to make it clear to the readers.

Tables and figures:

12. Figure 3 is difficult to understand. ‘Psoriasis + PURE 4’ and ‘Psoriasis + PsA’ seem not to be similar to one-another.

13. Please check for any dissimilarities between the calculations presented in the tables and the text in the manuscript.

14. Please update the footnotes for better understandability and explanation.

Reviewer #3:

1. The background introduction of PURE-4 is too brief. Could you provide a more detailed introduction?

2. Could you list the detailed items of the PURE-4 questionnaire?

3. It is noted that 219 participants were included in the study, but the final ROC curve was plotted with n=217. Could you explain why two people were excluded?

4. Could the ROC curve be included in the manuscript?

5. The baseline data for psoriatic arthritis are too sparse. Could some interesting indicators be added to enrich the content?

6. The manuscript mainly describes the specificity and sensitivity of PURE-4 in predicting the development of psoriatic arthritis one year later. Compared with other predictive indicators, does it have an advantage?

**Do you want your identity to be public for this peer review?** For information about this choice, including consent withdrawal, please see our Privacy Policy

Reviewer #1: No

Reviewer #2: No

Reviewer #3: **Yes:** Bing Wang

---

## [Author Response · Author response to Decision Letter 1]

3 Jun 2025

Dear editor,

As required in your previous email, we have uploaded an English translation document of all the original ethics documents provided previously in Spanish (also included).

To clarify the site-specific day, month, and year of the start and end of the recruitment period of the study, a table is included in the Supporting information section with all the required data. The new table has been referenced in the manuscript in the Methods section.

I thank you in advance for your help and look forward to your response.

Best regards,

---

## [Decision Letter · Decision Letter 1]

15 Oct 2025

Dear Dr. Queiro,

Thank you for submitting your manuscript to PLOS ONE. After careful consideration, we feel that it has merit but does not fully meet PLOS ONE’s publication criteria as it currently stands. Therefore, we invite you to submit a revised version of the manuscript that addresses the points raised during the review process.

**ACADEMIC EDITOR:  Title and Abstract**

- Briefly describe psoriatic arthritis as a disease.

- Briefly give the key findings of your study. Include key numeric data (including confidence intervals or *p*  values)

**Introduction**

- Briefly describe psoriatic arthritis as a disease.

- The gaps in knowledge and rationale for the study need to be mentioned.

**Methods**

- Please outline this section following the STROBE guidelines.

- You need to state in the Methods section that you have followed relevant EQUATOR guidelines. e.g., STROBE: “The reporting of this study conforms to STROBE guidelines”. (Insert new reference number, Note, you will probably need to renumber your references after this addition).

- Need subheading for more clarity

- How were the patients selected (e.g., consecutively, randomly, or selectively)?

- Describe how the study size was arrived at.

**Discussion**

- In the grand scheme of things, please attempt to summarize key findings in relation to the study's objectives.

- Have you compared your results with relevant previous papers, and cited those papers?

- Have you discussed the relevance and novelty of your study and what it adds to literature?

- Ensure there are the following Declarations sections at the end of your manuscript: Acknowledgements, Author contributions, Funding, Availability of data and materials statement, competing interests, Ethics approval and consent to participate.

We look forward to receiving your revised manuscript.

Kind regards,

Wesam Gouda, MD,PhD

Academic Editor

PLOS ONE

Journal Requirements:

Reviewers' comments:

Reviewer's Responses to Questions

**Comments to the Author**

Reviewer #1: (No Response)

Reviewer #2: (No Response)

Reviewer #3: (No Response)

2. Is the manuscript technically sound, and do the data support the conclusions?

Reviewer #1: Yes

Reviewer #2: Partly

Reviewer #3: (No Response)

3. Has the statistical analysis been performed appropriately and rigorously?

Reviewer #1: Yes

Reviewer #2: Yes

Reviewer #3: (No Response)

4. Have the authors made all data underlying the findings in their manuscript fully available?

Reviewer #1: No

Reviewer #2: No

Reviewer #3: (No Response)

5. Is the manuscript presented in an intelligible fashion and written in standard English?

Reviewer #1: Yes

Reviewer #2: Yes

Reviewer #3: (No Response)

Reviewer #1: Your study provides valuable evidence on the potential of the PURE-4 questionnaire to identify psoriasis patients at increased risk of developing PsA. However, several aspects require revision to strengthen the manuscript:

1. Statistical power: Please emphasize the small number of PsA cases (n=12 at one year) and interpret sensitivity/specificity estimates with caution, highlighting the wide confidence intervals.

2. Treatment confounding: Discuss more explicitly how systemic and biologic therapies might have influenced PsA onset and questionnaire performance.

3. Comparison with existing tools: Include a discussion of how PURE-4 compares with other screening questionnaires (e.g., EARP), addressing advantages and limitations.

4. Study design limitation: Acknowledge that administering the questionnaire only once precludes assessment of reproducibility or changes in score over time.

Addressing these points will improve the clarity, robustness, and overall impact of your work.

Reviewer #2: The study objectives are interesting and relevant for the early detection of psoriatic arthritis. The article is carefully written and methodology is robust. I have provided my comments below.

Abstract:

1. Please restructure the ‘materials and methods’ part in the abstract in order to clarify to the readers that the patients with psoriasis (without previous diagnosis of PsA) from Assessment I took part in Assessment II. Please distinguish which cross-sectional studies the assessments are related to. Information on the sensitivity and specificity analysis will improve the results section further.

Introduction:

2. As the validation study of the 'PURE-4 scale' was conducted in a sample from France between 2012 and 2014, the following sentence needs to be updated: “Although there are currently various tools for screening for PsA, neither a development methodology nor validation in routine clinical practice in Spain has been carried out at all times [11].” I understand that you indicated its necessity for the Spanish sample. Please make it clear for the reader. (Page 4, line 79)

3. In the Introduction part, in addition to the prevalence data, please add a short description on the disease and how early detection helps in reducing disease burden. I recommend expanding the literature review a little bit and include more recent studies on this topic.

4. I also recommend shortly reviewing the previous studies and describing a clear research gap.

Methods:

5. Which software was used to calculate the performance of the PURE-4 questionnaire for the early detection of potential PsA in terms of sensitivity, specificity, positive predictive value, negative predictive value? (Page 6, line 132). Discuss in brief what the term ‘area under the receiver-operating characteristic (ROC) curve’ indicates about the quality of the questionnaire.

6. Please refer to the Youden index and describe it briefly.

Results:

7. The numbers presented in the results section (Page 7, line 165) is different from what is presented in Figure 1. For example, the figure shows that Dx: no PsA (n = 223) and Dx: not evaluable (n= 11) participated in Assessment II. Shouldn’t be it 234 instead of 219? Please present identical values and check for any mistakes. In figure 3, please present the information on the participants who were not evaluable.

8. There are differences in the values between Table 1 and S2 Table. Please check for any mistakes. If necessary, please update the title of the tables.

9. How Dactylitis, Enthesitis, inflammatory back pain, peripheral joint pain, DLQI, etc were evaluated? Please describe in short in the methods section.

Discussion:

10. What were the sensitivity and specificity of the original validation study for the PURE-4 scale? Please compare with your results.

11. Please discuss which relevant questions of the questionnaire were answered one year before the PsA were identified during assessment II. How does it determine the performance of PURE-4 questionnaire for early detection of PsA? If Figure 4 includes data only from assessment II (e.g., 91.7% participants with PsA diagnosis had peripheral joint pain with swelling), how can it be stated that it has assisted in early detection of PsA if it is done during the same assessment? You presented related results (both from two assessments) in S2 Fig. Please update the discussion to make it clear to the readers.

Tables and figures:

12. Figure 3 is difficult to understand. ‘Psoriasis + PURE 4’ and ‘Psoriasis + PsA’ seem not to be similar to one-another.

13. Please check for any dissimilarities between the calculations presented in the tables and the text in the manuscript.

14. Please update the footnotes for better understandability and explanation.

Reviewer #3:

1. The background introduction of PURE-4 is too brief. Could you provide a more detailed introduction?

2. Could you list the detailed items of the PURE-4 questionnaire?

3. It is noted that 219 participants were included in the study, but the final ROC curve was plotted with n=217. Could you explain why two people were excluded?

4. Could the ROC curve be included in the manuscript?

5. The baseline data for psoriatic arthritis are too sparse. Could some interesting indicators be added to enrich the content?

6. The manuscript mainly describes the specificity and sensitivity of PURE-4 in predicting the development of psoriatic arthritis one year later. Compared with other predictive indicators, does it have an advantage?

**Do you want your identity to be public for this peer review?** For information about this choice, including consent withdrawal, please see our Privacy Policy

Reviewer #1: No

Reviewer #2: No

Reviewer #3: **Yes:** Bing Wang

---

## [Author Response · Author response to Decision Letter 2]

24 Nov 2025

Thank you for your letter and constructive comments concerning our manuscript entitled ‘Validation of performance of Spanish version of PURE-4 questionnaire for early identification of psoriatic arthritis after 1 year of follow-up in patients with psoriasis'.

We have carefully reviewed your comments and have revised the manuscript accordingly. Our responses are given in blue in a point-by-point manner in the Answers to reviewers comments_PURE-4 questionnaire document.

Changes to the documents are implemented in “track-changes” and we have submitted two revised versions, one with tracked-changes and one clean version without tracked-changes, for ease of your review.

We trust that the revised version is now suitable for publication and look forward to your feedback in due course.

Yours sincerely,

Rubén Queiro-Silva

---

## [Decision Letter · Decision Letter 2]

25 Jan 2026

Validation of performance of Spanish version of PURE-4 questionnaire for early identification of psoriatic arthritis after 1 year of follow-up in patients with psoriasis

PONE-D-24-54846R2

Dear Dr. Queiro,

We’re pleased to inform you that your manuscript has been judged scientifically suitable for publication and will be formally accepted for publication once it meets all outstanding technical requirements.

Kind regards,

Wesam Gouda, MD,PhD

Academic Editor

PLOS One

Additional Editor Comments (optional):

Reviewers' comments:

Reviewer's Responses to Questions

**Comments to the Author**

Reviewer #1: All comments have been addressed

Reviewer #2: All comments have been addressed

2. Is the manuscript technically sound, and do the data support the conclusions?

Reviewer #1: Yes

Reviewer #2: Yes

3. Has the statistical analysis been performed appropriately and rigorously?

Reviewer #1: Yes

Reviewer #2: Yes

4. Have the authors made all data underlying the findings in their manuscript fully available?

Reviewer #1: Yes

Reviewer #2: (No Response)

5. Is the manuscript presented in an intelligible fashion and written in standard English?

Reviewer #1: Yes

Reviewer #2: Yes

Reviewer #1: It is my pleasure to inform you that your revised manuscript, PONE-D-24-54846R2, 'Validation of performance of Spanish version of PURE-4 questionnaire for early identification of psoriatic arthritis after 1 year of follow-up in patients with psoriasis', has been accepted for publication in PLOS ONE. We appreciate the excellent quality of the revisions and the comprehensive way you addressed the feedback. Congratulations on this significant work.

Reviewer #2: (No Response)

**Do you want your identity to be public for this peer review?** For information about this choice, including consent withdrawal, please see our Privacy Policy

Reviewer #1: No

Reviewer #2: **Yes:** Dr. Ruhul Amin

---

## [Editor Report · Acceptance letter]

PONE-D-24-54846R2

PLOS One

Dear Dr. Queiro,

I'm pleased to inform you that your manuscript has been deemed suitable for publication in PLOS One. Congratulations! Your manuscript is now being handed over to our production team.

Kind regards,

on behalf of

Dr. Wesam Gouda

Academic Editor

PLOS One